# Impact of Albumin Binding Function on Pharmacokinetics and Pharmacodynamics of Furosemide

**DOI:** 10.3390/medicina58121780

**Published:** 2022-12-02

**Authors:** Gerd Klinkmann, Sebastian Klammt, Malte Jäschke, Jörg Henschel, Martin Gloger, Daniel A. Reuter, Steffen Mitzner

**Affiliations:** 1Department of Anaesthesiology, Intensive Care Medicine and Pain Therapy, University Medical Center Rostock, Schillingallee 35, 18057 Rostock, Germany; 2Department of Internal Medicine, University Medical Center Rostock, Schillingallee 35, 18057 Rostock, Germany; 3Institute for Diagnostic and Interventional Radiology, University Medical Center Rostock, Schillingallee 35, 18057 Rostock, Germany; 4Division of Nephrology, Department of Internal Medicine, Medical Faculty, University of Rostock, Ernst-Heydemann-Str. 6, 18057 Rostock, Germany; 5Department of Extracorporeal Immunomodulation, Fraunhofer Institute for Cell Therapy and Immunology, Schillingallee 68, 18057 Rostock, Germany

**Keywords:** albumin binding capacity, critical care, de-resuscitation, diuretics, fluid accumulation, fluid overload, fluid removal, furosemide, intensive care, loop diuretics

## Abstract

*Background and Objectives*: Albumin binding of the loop diuretic furosemide forms the basis for its transport to the kidney and subsequent tubular secretion, which is a prerequisite for its therapeutic effects. Accordingly, high albumin concentrations should result in higher efficacy of furosemide. However, study results on the combination of furosemide in conjunction with albumin, and on the efficacy of furosemide in hypoalbuminemia, did not confirm this hypothesis. The aim of this study was to determine the efficacy of furosemide not only in relation to albumin concentration, but also taking albumin function into account. *Materials and Methods*: In a prospective and non-interventional clinical observational trial, blood and urine samples from 50 intensive care patients receiving continuous intravenous furosemide therapy were evaluated. Albumin binding capacity (ABiC) determination allowed conclusions to be drawn about the binding site-specific loading state of albumin, by quantifying the unbound fraction of the fluorescent marker dansylsarcosine. In addition, assessment of the total concentration of furosemide in plasma and urine, as well as the concentration of free furosemide fraction in plasma, was performed by HPLC–MS. The efficacy of furosemide was evaluated by the ratio of urine excretion to fluid intake. *Results*: In patients with an ABiC ≥ 60% free furosemide fraction was significantly lower compared to patients with a lower ABiC (*p* < 0.001), urinary furosemide concentration was higher (*p* = 0.136), and a significantly higher proportion of infused furosemide was excreted renally (*p* = 0.010). ABiC was positively correlated (r = 0.908, *p* = 0.017) with increase in the urine excretion to fluid input ratio after initiation of furosemide therapy. *Conclusions*: ABiC could serve as a marker for individual response to furosemide and could be used to generate patient-specific therapeutic regimens. In view of the relatively low number of patients in this study, the relationship between furosemide efficacy and albumin function should be investigated in larger studies in the future.

## 1. Introduction

### Background

Loop diuretics represent an important therapeutic option for patients with fluid retention. Via inhibition of the sodium-potassium-2-chloride ion transporter in the thick ascending portion of the loop of Henle, they prevent reabsorption of 20–30% of the primary filtered sodium. The ability of the kidney to concentrate primary urine by osmotic forces is significantly reduced and, consequently, water excretion is increased [1].

The most commonly used loop diuretic is furosemide [2]. Furosemide is a potent, short-acting diuretic, commonly used for the treatment of edema of cardiac, hepatic or renal origin. Besides the prescription for mobilization of excess water and sodium, furosemide is part of the care for different clinical settings, such as control of hypertension, ascites, pulmonary edema, and symptomatic hypercalcemia. It is poorly metabolized and eliminated in urine either unmodified or as glucuronide conjugate. Furosemide binds reversibly to albumin. Albumin is the most highly concentrated plasma protein in humans and, in addition to transporting various exogenous and endogenous ligands, performs other essential functions, such as maintaining colloid osmotic pressure [3].

Furosemide binds mainly to binding site I of the albumin molecule [4,5,6]. In this process, the furosemide–albumin complex is stabilized by the formation of hydrogen bonds, as well as by hydrophobic interactions [6]. In addition to the binding site in subdomain IIa (binding site I), furosemide binds to other amino acid residues of the albumin molecule. Earlier data suggested a second binding site, also located in subdomain IIa [5], but more recent data hints at binding to amino acid residues, some of which are also part of Sudlow’s binding site II. Binding to these residues occurs with lower affinity than to binding site I [6]. The loading state of binding site II is determined by ABiC, which may provide valuable information about the binding behavior of the entire albumin molecule.

Due to the large number of ligands and binding sites of the albumin molecule, numerous interactions are possible: drugs can compete with other drugs or their metabolites for binding sites, but endogenous ligands can also be involved in the interactions. In this context, the interactions occur not only via competitive displacement reactions, but also via allosteric conformational changes of the albumin molecule caused by ligand binding. Furthermore, electrostatic interactions also influence the binding of ligands to albumin. The aforementioned interactions may become clinically relevant when the change in free amount of a drug is affected, due to interaction with other ligands [4,7,8].

Furosemide normally binds to albumin at approximately 95–99% [5,6]. Due to this high protein binding, even slight changes in binding to albumin can influence the pharmacologically active free fraction of furosemide and, thus, its efficacy. Hence, less albumin binding may result in a higher free furosemide fraction in the plasma and less furosemide in the urine. It has already been shown that both fatty acids and toxins accumulating in uremia have an influence on the binding between furosemide and albumin [4,6,9].

The interactions presented suggest that the effect of furosemide is largely dependent on albumin-coupled transport into the kidney. Consequently, when patients do not respond to therapy with furosemide, concomitant administration of albumin to improve the transport situation for furosemide seems to be a possible solution to the problem. However, the results of several studies in patients with albumin deficiency, who were given furosemide and additionally albumin, were not consistent and, thus, could not confirm this hypothesis [10,11,12,13,14].

To assess the influence of albumin on the effect of furosemide, not only the total albumin plasma concentration should be considered, but also albumin function could be a meaningful parameter to explain the different responses of patients to therapy with furosemide.

Within the framework of a prospective and non-interventional observational clinical study, we aimed to test the hypothesis of albumin function affecting the efficacy of furosemide.

## 2. Material and Methods

### 2.1. Design, Settings and Study Population

This clinical observational study was conducted on patients who received continuous intravenous furosemide therapy as part of intensive care treatment and who met the inclusion criteria. In addition to continuous intravenous furosemide therapy for at least five hours, patients were required to be at least 18 years of age and to provide written informed consent. Exclusion criteria were known chronic renal insufficiency requiring dialysis, pregnancy and lactation, as well as the following events during the balancing interval: surgery, examinations with contrast medium administration and/or the use of renal replacement procedures.

Blood samples were taken from patients that were on continuous intravenous furosemide therapy for at least five hours. To assess the efficacy of furosemide, the ratio of the amount of urine excreted to the amount of fluid input over an equivalent period of time was determined. For this purpose, fluid intake and fluid output were balanced over 24 h. This balancing interval began at the same time as blood collection and ended after 24 h. Finally, a urine sample was taken from the urine collected over the balancing interval. Figure 1 provides the timeline of the study protocol.

### 2.2. Measurement of Albumin Concentration

Determination of albumin concentration is a prerequisite for determination of albumin binding capacity (ABiC). A colorimetric assay was used to measure albumin concentration. Measurements were performed on the CobasR Mira Plus automated analyzer (Hoffmann-La Roche AG, Basel, Switzerland) using the reagent “ALBUMIN Bromcresol Green” (Labor + Technik Eberhard Lehmann GmbH, Berlin, Germany).

### 2.3. Estimation of ABiC

ABiC was determined by an indirect method, based on the estimation of the unbound fraction of a specific albumin-bound marker in a plasma sample. By comparison with the fraction of the unbound marker in a reference albumin solution, the site-specific binding capacity of the sample was expressed semiquantitatively. Plasma samples were diluted to a concentration of 150 µmol/L albumin and incubated with an albumin binding site II-specific fluorescent marker [dansylsarcosine (DS), 150 µmol/L]. The albumin-free filtrate was obtained in a separation step (Centrisart I, 20,000D; Sartorius GmbH, Göttingen, Germany), and fluorescence was measured after addition of human serum albumin (300 µmol/L) as fluorescence enhancer (Fluoroscan; Thermo Labsystems, 355/465 nm). In parallel, the same procedure was performed with standard albumin as reference. A standardized virus-inactivated human serum preparation from pooled human plasma (Biseko^®^; Biotest Pharma GmbH, Dreieich, Germany) was used as a reference for ABiC. The binding capacity for the marker was quantified according to the following equation:$$\mathrm{ABiC}\;[\%]=\frac{\mathrm{fluorescence}\;\mathrm{in}\;\mathrm{ultrafiltrat}\;\mathrm{of}\;\mathrm{the}\;\mathrm{reference}\;({\mathrm{Biseko}}^{®})}{\mathrm{fluorescence}\;\mathrm{in}\;\mathrm{ultrafiltrat}\;\mathrm{of}\;\mathrm{the}\;\mathrm{sample}}\;\times \;100$$

Figure 2 illustrates a schematic overview of the ABiC determination. Albumin binding capacity (ABiC) determination allowed conclusions to be drawn about the binding site-specific loading state of albumin by quantifying the unbound fraction of the fluorescent marker dansylsarcosine.

### 2.4. Furosemide Extraction

Furosemide had to be transferred from the patients’ blood and urine samples into a solution separated from proteins as HPLC–MS common medium. This step was necessary, among others, to prevent clogging of the ion source of the mass spectrometer. The basis of this method was protein precipitation by chemical denaturation followed by liquid–liquid extraction of the analyte [15]. Given its structurally similar properties to furosemide, diclofenac was chosen as the internal standard. For the plasma samples, 20 μL of a diclofenac–methanol solution (diclofenac concentration 300 μg/mL) served as the internal standard, and, for the urine samples, 40 μL of the same solution was used as the internal standard. An amount of 300 μL of the plasma or urine sample was added to the internal standard. Subsequently, mixing was performed for 20 s in a vortex mixer. Protein denaturation was achieved by adding 450 μL of acetonitrile (ACN) followed by incubation at room temperature in aThermoMixerR at 1200 rpm for 20 min. Addition of 600 μL ethyl acetate, followed by mixing in a vortex mixer for two minutes, extracted the analytes. Centrifugation (Eppendorf Centrifuge 5810R, Eppendorf Vertrieb Deutschland GmbH, Wesseling-Berzdorf, Germany) at 14,000 rpm for 10 min was used to separate the denatured proteins. The organic phase was transferred to a 2.0 mL Eppendorf TubeR and the solvent was evaporated in a Savant DNA 120-SpeedVacR vacuum concentrator (Thermo Fisher Scientific Inc., Waltham, MA, USA) at medium manual drying rate for 90 min. Then, the sample was taken up with 300 μL of a methanol–water mixture (MeOH/H2O = 70:30), homogenized with the vortex mixer for one minute and, finally, centrifuged at 12,000 rpm for three minutes. For HPLC, 150 μL of the supernatant was transferred to appropriate HPLC sample tubes (Amchro GmbH, Hattersheim, Germany). Plasma and urine blank samples were used for calibration and quality control, to which either 20 μL of quality control solution or 20 μL of calibration solution were added. In order to determine not only the total concentration of furosemide, but also the free fraction not bound to albumin, in addition to the extraction process described, the free furosemide was separated from each plasma sample by ultrafiltration and measured separately. For this purpose, 300 μL of each plasma sample was transferred to CentrisartR I ultrafiltration units (Sartorius, Gottingen, Germany), with a molecular weight cut off of 20,000 Da, and centrifuged at 2840 G for 60 min before the addition of ACN.

### 2.5. High Performance Liquid Chromatography

A YMC-Triart C18 ExRS pre- and separation-column (YMC Europe GmbH, Dinslaken, Germany) was used to separate the serum and urine samples. Reversed-phase chromatography, which applies a nonpolar or only weakly polar stationary phase and a strong polar solvent, was used according to the manufacturer’s specifications. After sample separation by HPLC, mass spectrometric detection was performed.

### 2.6. Mass Spectrometry

Mass spectrometric analysis was performed with an LCMS-2020 Single Quadropole Mass Spectrometer (Shimadzu Deutschland GmbH, Duisburg, Germany), according to the manufacturer’s specifications.

### 2.7. Ethics Approval, Consent to Participate

This study was approved by the local ethics committee (Reg. No.: A-2014-0119) and written informed consent was obtained from all patients and volunteers.

### 2.8. Statistical Analysis

Statistical analysis of the data was performed using IBM SPSS Statistics (version 27, Chicago, IL, USA). The results were expressed as the mean ± standard deviation (SD) and range. Box plots were used for graphics. The horizontal line within the boxes represents the median, whereas the upper part represents the 75th, and the lower part the 25th, percentiles. The whiskers represent the range of the values, whereas the circles and the asterisks show the outliers (extreme values that deviated from the rest of the sample). According to the distribution of data (using Shapiro–Wilk test), Mann–Whitney U Test was used for two independent samples for continuous variables. Linear relationships between two interval-scaled, and normally distributed, characteristics were quantified using Pearson’s correlation coefficient (r). In the case of non-normally distributed interval-scaled characteristics, the Spearman rank correlation coefficient (rS) was calculated. Statistical differences were considered significant at a *p* value <0.05 and highly significant at *p* < 0.01. 

### 2.9. Study Population 

Table 1 provides demographic information about the study population.

Patients presented with different reasons for intensive care unit admission. A further classification of the patients could be evaluated, based on the indication for furosemide therapy. Presence of sepsis was the most common reason for intensive care treatment. The most common indication for continuous intravenous furosemide therapy was acute kidney injury, followed by acute on chronic renal failure. Renal failure was most frequently in association with sepsis. Acute kidney injury was diagnosed, based on the established KDIGO classification. Figure 3 provides an overview of the study population in terms of distribution of admission diagnosis and indication for furosemide therapy.

## 3. Results

### 3.1. ABiC and Indication for Furosemide Therapy

According to the different groups of patients identified and categorized, in terms of their indication for initiation of furosemide therapy, ABiC was determined. This was found to be reduced in all groups. ABiC was not significantly different in patients with acute kidney injury (64.0 ± 14.1), acute on chronic renal failure (65.4 ± 10.8), end-stage-renal-disease (ESRD) without the presence of acute kidney injury (72.5 ± 9.2), and in patients treated with furosemide without the presence of renal disease (67.2 ± 6.7) (*p* = 0.784). The “Other” category included patients treated with furosemide for pulmonary edema, leg edema, fluid de-resuscitation after sepsis, and acute respiratory distress syndrome (ARDS). These patients had no known renal comorbidities and were not suffering from acute kidney injury (Figure 4).

### 3.2. ABiC and GFR

At the beginning of the balancing interval ABiC and GFR showed a highly significant correlation (rS = 0.404; *p* = 0.003; *n* = 44).

### 3.3. ABiC and Albumin Concentration

There was a highly significant moderately positive correlation (r = 0.546; *p* < 0.001; *n* = 49) between ABiC and albumin concentration.

### 3.4. Free Furosemide Fraction

The inverse correlation between ABiC and free furosemide fraction (*p* < 0.001, rS = −0.638) was stronger than the correlation between albumin concentration and free furosemide fraction (*p* < 0.001, rS = −0.494). There was no significant correlation between the amount of furosemide infused within five hours before blood sampling and the free furosemide fraction (*p* = 0.062, rS = −0.231). Patients in whom ABiC was less than 60% had a highly significant higher free furosemide fraction than patients in whom ABiC was at least 60% (*p* < 0.001) (Figure 5). Free furosemide fraction was not significantly different in patients who received a total of 100 mg of furosemide within five hours prior to blood collection, compared to patients who were infused with a lower amount of furosemide, 53.44 ± 8.26 mg [42.5–75 mg] (*p* = 0.105). Consideration of free furosemide fraction in relation to the indication for furosemide therapy showed that free furosemide fraction was lowest in patients without renal disease.

### 3.5. Furosemide Concentration in Urine

Furosemide therapy was terminated or interrupted in 15 of the 50 patients before the end of the balancing interval. Of the remaining 35 patients, 19 patients received 20 mg furosemide per hour without a dose change and, thus, a total of 480 mg furosemide over the balancing interval. Two patients were treated constantly with 10 mg furosemide per hour, one patient with 5 mg furosemide per hour. In 13 patients, the furosemide dosage was changed during the balancing interval. The amount of furosemide infused over 24 h in these patients was 249.6 ± 119.0 mg [102–438 mg].

ABiC was significantly weakly correlated with urinary furosemide concentration (*p* = 0.005; rS = 0.482). In the subgroup infused with 480 mg furosemide over the balancing interval, there was a moderately significant correlation of ABiC and urinary furosemide concentration (*p* = 0.041; rS = 0.463).

In patients in whom ABiC was at least 60%, urinary furosemide concentration did not significantly differ (*p* = 0.136) from patients in whom ABiC was less than 60%. The amount of furosemide infused was not significantly different in the two groups, but was lower, on average, in patients with an ABiC of at least 60%, than in patients with an ABiC below 60% (*p* = 0.237).

### 3.6. Amount of Furosemide Excreted

In patients in whom the ABiC was at least 60%, the ratio of the amount of furosemide excreted in the urine to the amount of furosemide infused over the balancing interval was significantly higher than in patients in whom the ABiC was less than 60% (*p* = 0.010) (Figure 6).

### 3.7. Efficacy of Furosemide

The ratio of the amount of urine excreted during the balancing interval to the amount of fluid received over the same period was used to assess the efficacy of furosemide. The ratio of urine excretion to fluid intake was not significantly different in patients with an ABiC of at least 60% from patients with a lower ABiC (*p* = 0.759).

### 3.8. ABiC and Response to Furosemide

To assess response to furosemide, the ratio of urine output to fluid intake during the balancing interval was compared with the 24 h before initiation of furosemide therapy. For this purpose, the ratio of urine excretion to fluid intake from the 24 h before the start of furosemide therapy was subtracted from the same ratio during the balancing interval. Positive values, thus, represented an increase in the ratio of urine output to fluid input after the start of furosemide therapy. Only patients were considered in whom initiation of continuous intravenous furosemide therapy did not occur more than 24 h before blood collection. There was a significant correlation (r = 0.908; *p* = 0.017; *n* = 5) of ABiC with increase in urine output to fluid intake ratio after initiation of furosemide therapy (Figure 7).

### 3.9. Album Substitution

Seven patients received human albumin preparations during the balancing interval. Three patients were treated with “Humanalbumin Biotest 20%” (Biotest Pharma GmbH, Dreieich, Germany), two patients with “Humanalbumin 20% Behring, low salt” (CSL Behring GmbH, Marburg, Germany) and one patient with “Humanalbumin 20% Octalbin” (Octapharma GmbH, Langenfeld, Germany). In one patient, the human albumin preparation was unspecified. The ratio of urine output to fluid intake was not significantly different in patients with albumin substitution during the balancing interval compared to the rest of the study population (*p* = 0.643). The amount of furosemide infused over the balancing interval was not significantly different in either group (*p* = 0.643).

### 3.10. Summary

In this observational clinical study, the efficacy of furosemide, as measured by the ratio of urine output to fluid intake, was examined for the first time as a function of ABiC. For this purpose, data from 50 patients treated with furosemide during intensive care were analyzed. When compared to patients with a lower ABiC, patients with an ABiC of at least 60% had a highly significant lower free furosemide fraction, a higher urinary furosemide concentration, and a significantly higher proportion of infused furosemide excreted in the urine. However, the ratio of urinary excretion to fluid infusion was not significantly different in the two groups. Higher ABiC was associated with better response to initiation of furosemide therapy. Moreover, to the best of our knowledge, this study was the first to demonstrate that ABiC is also reduced in patients suffering from acute kidney injury.

## 4. Discussion

Several studies have already shown a correlation of ABiC with the clinical course of various diseases. Thus, ABiC is decreased in patients with chronic renal failure. The deterioration of ABiC is negatively correlated with the serum concentration of indoxyl sulfate, a toxin that accumulates in uremia and is present in the plasma, primarily in albumin-bound form. In addition, ABiC has been shown to be strongly correlated with the level of glomerular filtration rate [16]. Further, ABiC in patients with decompensated cirrhosis is negatively correlated with disease severity, as measured by Child–Turcotte–Pugh (CTP) classification and the model for end-stage liver disease (MELD). Interestingly, however, no correlation was found between albumin level and ABiC [17].

Klammt et al. reported an ABiC of 102.6 ± 12% in 12 healthy subjects [18]. ABiC was significantly lower in our study populations at 65.2 ± 12.2%. Similarly low values of ABiC (median: 63%) were found for patients with decompensated liver cirrhosis [17]. For patients with chronic renal failure, Klammt et al. described a high positive correlation (rS = 0.881; *p* < 0.001; *n* = 104) between ABiC and GFR [16]. Our data yielded a low positive correlation (rS = 0.404; *p* = 0.003; *n* = 44) of GFR with ABiC. However, the study population was smaller and more heterogeneous. In addition to patients with acute, or acute on chronic renal failure, we included patients without the presence of renal disease. Klammt et al. merely examined patients with chronic renal insufficiency, without the presence of acute renal failure, who also did not require intensive medical care; thus, a less critical health status seemed to be assumed [16]. Nevertheless, the hypothesis that there was a decrease in ABiC with decrease in GFR was also supported by our data. 

The hypothesis of patients’ critical health status being associated with decreased ABiC was also supported by evidence that ABiC did not differ significantly in patients with acute renal failure, acute on chronic renal failure, ESRD without the presence of acute renal failure, and in patients treated with furosemide without the presence of renal disease. Although impairments in ABiC are known for patients with decompensated liver cirrhosis, in addition to patients with chronic renal failure [16,17], large portions of the study population did not have any of these conditions and still showed decreased ABiC. Notably, our data provided preliminary evidence that, in addition to chronic renal failure, acute renal failure might also be associated with a reduction in ABiC. However, it is unclear whether acute renal failure, or the overall critical condition of the patients, caused the decrease in ABiC. In patients with acute deterioration of chronic liver disease, significantly higher ABiC was found in survivors compared with patients who died during the study period [19]. Although our data did not show significant differences in survivors and non-survivors, ABiC was significantly higher in patients who did not require mechanical ventilation. This might also indicate that ABiC was reduced in critically ill patients. 

The decrease in ABiC, with the increase in the number of drugs used, was not very surprising; after all, each drug increases the likelihood of occupying binding site II. Finally, ABiC was inversely correlated with the number of drugs used.

Inoue et al. revealed that albumin binding is a prerequisite for the transport of furosemide into the kidney [20]. Accordingly, the highest possible level of bound furosemide and, thus, a low free furosemide fraction ought to be fundamental for a profound effect of furosemide. This thesis was also supported by the results of this work. Plasma concentration of free furosemide fraction, in contrast to total furosemide concentration, showed no significant correlations with urine concentration of furosemide and the ratio of excreted to infused furosemide. In order to ensure comparability of data among the study patients we determined, in addition to the absolute concentration of free furosemide, also the proportion of the free fraction to total furosemide concentration, i.e., the free furosemide fraction. While it could be assumed that different furosemide doses occurring in the study population influenced the absolute concentrations of the free and bound fractions, the ratio of free to bound fraction should remain unaffected by the amount of furosemide infused. This hypothesis was supported by the fact that there was no significant correlation between the free furosemide fraction and the amount of furosemide infused within five hours before blood sampling. 

Although overall there was only a moderate negative correlation between ABiC and free furosemide fraction, ABiC was, nevertheless, more negatively correlated with free furosemide fraction than albumin concentration. This was noteworthy, because ABiC describes the binding site-specific loading state of albumin binding site II, but furosemide binds mainly to the binding site I [5,6,21]. This suggested that, although only the loading state of binding site II was determined, ABiC could still provide valuable information about the binding behavior of the entire albumin molecule and be used as a possible “global” marker of albumin function. It was also possible that the low affinity binding of furosemide to amino acid residues, partially belonging to binding site II, could explain the correlation with ABiC, as initially described by Zaidi et al. [6].

It was shown that patients with an ABiC of at least 60% had higher urinary furosemide concentrations than patients with an ABiC below 60%, although the latter patients were infused with more furosemide, on average, over the balancing interval. Thus, a higher ABiC appeared to allow more effective transport of furosemide to its site of action.

The limited efficacy of furosemide in our study population might have had several causes. In patients with chronic renal failure, tubular secretion of diuretics is limited by accumulating substances, such as urates, and the possible presence of metabolic acidosis [22]. In patients with heart failure and patients with liver cirrhosis, tubular secretion of diuretics is known to be limited by renal vasoconstriction, due to decreased cardiac output or vasodilation in the splanchnic area [23]. Furthermore, it is known that furosemide clearance can be severely limited in acute renal failure [24]. These mechanisms might have been responsible for the fact that the renally excreted fraction of furosemide in this study was only about 19% on average, well below the expected rate of two-thirds [2]. However, they do not explain why the excreted proportion of furosemide was not as strongly correlated with urinary excretion as expected. Given the acute tubular necrosis that may be present in acute renal failure, one explanation could be that no physiological response to furosemide therapy could be expected. Acute tubular necrosis is a major trigger of intrarenal renal failure in patients treated in intensive care. In most patients, acute renal failure occurred in the setting of sepsis. Gomez and Kellum emphasized that sepsis-induced acute renal failure could not be explained solely by renal hypoperfusion and the occurrence of acute tubular necrosis, and other mechanisms, such as microvascular dysfunction and inflammatory responses affecting tubular cells, might be of higher pathophysiological significance than acute tubular necrosis [25]. It is possible that these mechanisms explain the reduced efficacy of the furosemide fraction reaching the kidney, and, thus, demonstrating why patients with an ABiC of at least 60% excreted a significantly higher fraction of infused furosemide renally than patients with lower ABiC but did not excrete more urine relative to fluid input.

In this work, furosemide was not shown to be more effective in patients who received albumin supplements during the balancing interval compared with patients without albumin supplementation. This was consistent with the poor correlation of albumin concentration with urine excretion to fluid intake ratio and also with the results of much larger studies. In a meta-analysis on the use of albumin to overcome diuretic resistance in a total of 343 hypoalbuminemic patients, Kitsios et al. found a significant increase in urine and sodium excretion with furosemide and albumin administration at eight hours, compared with furosemide alone, whereas no significant differences remained at 24 h [12]. Such short-lasting effects were not detectable with the study design used in this work, as the infusion duration of albumin preparations was significantly shorter than the balancing interval in all patients. However, it is also possible that there is simply no effect of albumin substitution on diuresis, which would fit with the results of other studies [13,14].

## 5. Conclusions

ABiC has the potential to be an important predictor of response to furosemide therapy. Thus, it could form a cornerstone for the use of personalized therapy concepts to optimize the outcome of patients treated with loop diuretics due to acute heart failure, as called for by Palazzuoli et al. [26]. For example, ABiC determination before therapy initiation would be conceivable to optimally estimate the necessary furosemide dosage. A possible consequence of low ABiC would be the administration of albumin preparations without stabilizers.

A similar concept would also be conceivable for patients treated with furosemide in the context of acute renal failure. Especially considering the critical indication for diuretic therapy in the presence of acute renal failure [27], ABiC, if the use of diuretics is targeted, could help to minimize the necessary dose on a patient-specific basis. This is particularly desirable because furosemide clearance may be significantly reduced in acute renal failure [24] and the use of high doses is thought to promote the occurrence of significant adverse side effects, such as ototoxic effects [28].

While limitations in ABiC are known for patients with chronic renal failure and decompensated liver cirrhosis, this study population showed severely reduced ABiC even without the presence of these conditions [16,17]. Of particular importance, this is the first report demonstrating ABiC to be reduced in patients suffering from acute kidney injury and, thus. representing a potential biomarker for screening. Further data collection is urgently needed in this regard.

The potential impact of critical health status and, in particular, the presence of sepsis on ABiC could be further investigated in the future. In particular, objectification of health status via, for example, the collection of the Sequential Organ Failure Assessment (SOFA) score, elevations of which are associated with increased mortality risk, would be desirable [29].

## Figures and Tables

**Figure 1 medicina-58-01780-f001:**
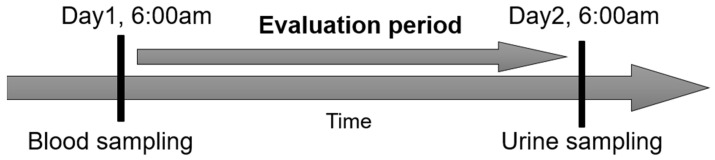
Timeline. Blood samples were collected from patients undergoing continuous intravenous furosemide therapy for at least five hours. Fluid intake and fluid output were determined over 24 h. The balancing interval began at the same time as the blood was taken and ended after 24 h. Finally, a urine sample was taken from the urine collected over the balancing interval.

**Figure 2 medicina-58-01780-f002:**
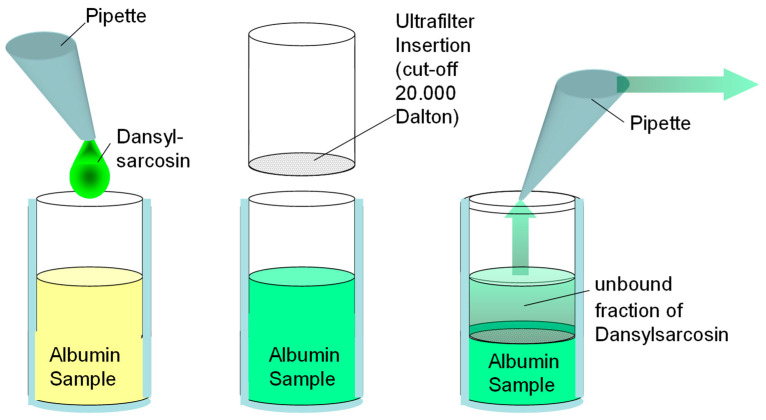
Albumin Binding Capacity (ABiC): a binding site II-specific test.

**Figure 3 medicina-58-01780-f003:**
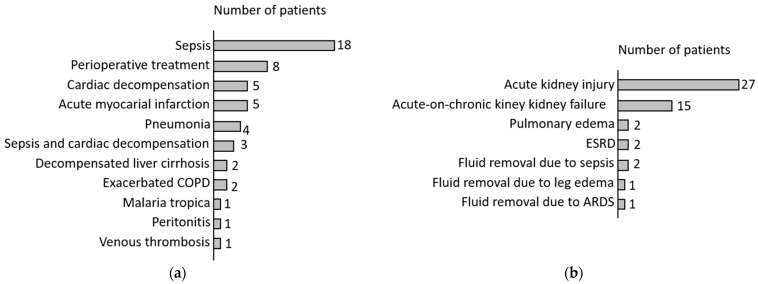
(**a**) Study population—Admission diagnosis; (**b**) Study population—Indications for Furosemide.

**Figure 4 medicina-58-01780-f004:**
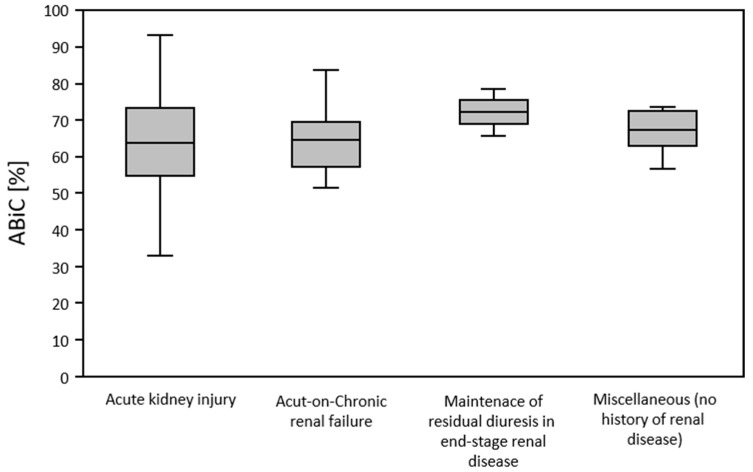
ABiC in dependence of the indication for furosemide therapy.

**Figure 5 medicina-58-01780-f005:**
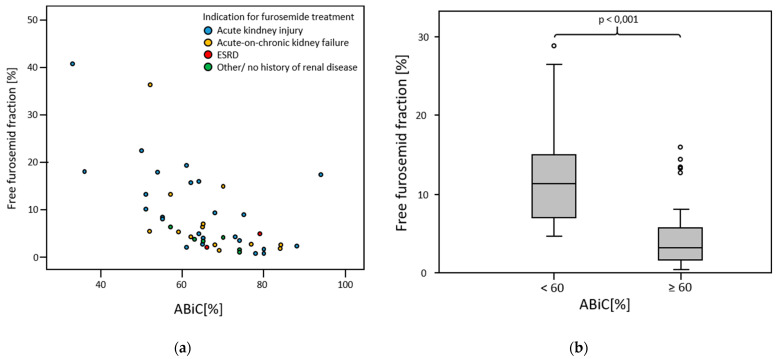
(**a**) Free furosemide fraction in relation to ABiC; The lower the ABiC, the higher the free fraction of furosemide. (**b**) Free furosemide fraction in relation to ABiC groups < 60% and ≥60%.

**Figure 6 medicina-58-01780-f006:**
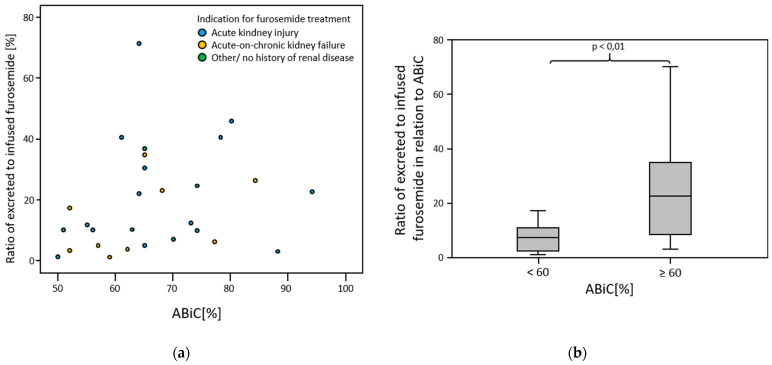
(**a**) Ratio of excreted to infused furosemide in relation to ABiC; (**b**) Ratio of excreted to infused furosemide in relation to ABiC groups < 60% and ≥60%.

**Figure 7 medicina-58-01780-f007:**
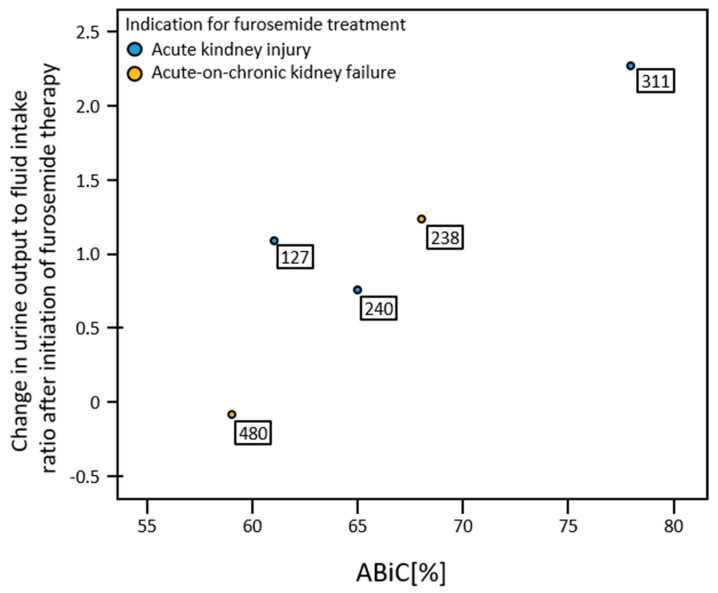
Change in urine output to fluid intake ratio after initiation of furosemide therapy. The boxes provide the cumulative amount of furosemide infused over the balancing interval [mg].

**Table 1 medicina-58-01780-t001:** Study population demographics.

Parameter	Mean	SD	Min.	Max.
Age [years]	67.5	10.3	46.0	84.0
Weight [kg]	89.2	23.4	40.0	156.0
BMI [kg/m^2^]	30.3	8.6	15.6	50.8

## Data Availability

The datasets used and analyzed during the current study are available from the corresponding author on reasonable request.

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
