# Peer review of "Impact of Albumin Binding Function on Pharmacokinetics and Pharmacodynamics of Furosemide"

_medicina, 2022, doi:10.3390/medicina58121780_

Round 1
Reviewer 1 Report
The manuscript presents interesting data from a prospective clinical trial on the fate of the “loop diuretic” Furosemide. It stimulates the curiosity of nephrologists and recommends further actions. Before considering the manuscript for publication in medicina some corrections should be performed by the authors.
1. Abstract
a) The authors should describe in the first line of the abstract furosemide as a “loop diuretic”
b) Line 19: Add “plasma level of “furosemide
c) Line 22: Add in relation to “albumin” concentration
d) Open question when talking about n=50 patients, what are the controls?
2. Introduction
a) The authors refer Line 92: the authors should to a non-interventional study and should add
that this was a “clinical“ trial.
b) This reviewer is missing a hypothesis to be tested in this study at the end of the introduction,
such as “We want to test the hypothesis that albumin function has an impact on furosemide
efficacy”
3. Material and Methods
a) Study population should be precisely addressed. N = 50? Controls?
b) Legend of Figure is insufficient and should describe in short words, what has been done. And
e.g, add the time frame between blood sampling and urine sampling, such as “ 24hrs”
c) Line 124 on page 3: Meaning of 150 lmol/L unclear
d) Figure 2 should have a description in its legend about the function of Dansylsarcosin
e) Line 148 on page 4: “incubating” should read “incubated”
4. Results
a) The whole description on Study population should be replaced into the Materials & method
section, i.e. from page 5, Line 195 to page 6, line 225.
b) Figure 4: please describe in legend both, what you observe, the abbreviation of ABiC and
whether your observations are or are not statistically different.
c) Consistency between n=50 and n= 49 on page 7, line 241, why difference?
d) Please describe and explain in legend what you see in Figure 5
e) Figure 7: please explain and refer to hypothesis what you see in figure 7
5. Discussion
a) First part of discussion should be replaced to “results”
6: Reference list: Please correct letter size in Ref 15.
Reviewer 2 Report
In this prospective observational study in adults on a furosemide infusion, patients with a higher albumin binding capacity had higher excretion of furosemide in the urine, though the efficacy of furosemide was not increased. This is a novel study that may have clinical utility to multiple different specialties.
Major revisions
1. For those less familiar with furosemide pharmacodynamics it may be helpful to clarify why less albumin binding and thus more free furosemide doesn’t result in greater furosemide in the urine. This is well stated in the discussion, but could be more clear in the introduction.
2. Furosemide is mainly bound to binding site I, but ABiC measures capacity at binding site II. It is not entirely clear in the introduction why measuring this may be helpful, though mentioned in the discussion.
3. I’m not clear how fluid input and output were balanced.
4. Was furosemide dose kept constant during the 24 hours?
5. Is there an r value for Figure 5a and for the correlations mentioned in lines 243-244?
6. Can we have correlation statistics for section 3.6?
7. Line 400: would soften the language here given the low number of patients that received albumin so it doesn’t sound like you are making conclusions based on an n of 7.
8. Line 417-419: This sentence sounds like a recommendation about administering albumin preparations but you have already sent time discussing that albumin administration doesn’t work, unless I’m misreading the intent of this sentence.
Minor revisions
1. The abbreviation ABiC is defined in the abstract but not in the main body of the paper.
2. The test used for correlation is not listed in the statistical methods.
3. How was acute renal failure defined?
4. Much of the demographics in the first paragraph of the results would be easier to read as a table.
